# A Conceptual Framework for the Co-Construction of Human–Dog Dyadic Relationship

**DOI:** 10.3390/ani15192875

**Published:** 2025-09-30

**Authors:** Laurie Martin, Colombe Otis, Bertrand Lussier, Eric Troncy

**Affiliations:** GREPAQ (Groupe de Recherche en Pharmacologie Animale du Québec), Faculty of Veterinary Medicine, Université de Montréal, Saint-Hyacinthe, QC J2S 2M2, Canada; laurie.martin.4@umontreal.ca (L.M.); colombe.otis@umontreal.ca (C.O.); bertrand.lussier@umontreal.ca (B.L.)

**Keywords:** owner, canine, self-determination, attachment, competence, education, autonomy, hybrid

## Abstract

The relationship and interactions within owner–dog dyads may be shaped by a dynamic process known as co-construction, wherein each individual influences and adapts to each other. Although this concept is mentioned in the literature, it has yet to be clearly defined or fully conceptualized. This narrative review explores how self-determination theory can enhance our understanding of this phenomenon. This review examines how both owners and dogs may fulfill their three fundamental psychological needs: competence, autonomy, and relatedness. It also explores various methodological approaches for studying co-construction, highlighting their interests and limitations. Co-construction unfolds over time and across varied contexts. The benefits are not immediate; its expression appears to be influenced by biopsychosocial, experiential, and environmental factors unique to each dyad and its individual members. While the current literature presents several proto-tools, drawing on both qualitative and quantitative methods to explore dyadic co-construction, none directly measures the process in a bidirectional way. The development of more integrative, interactive tools would allow for a deeper and more nuanced understanding of the co-construction process within owner–dog relationships.

## 1. Introduction

Co-construction generally refers to processes in which individuals jointly participate in the design, implementation, and execution of actions [1]. Although frequently cited, the concept remains under-defined and lacks a widely accepted theoretical framework [1]. When defined, it is typically framed as a combination of cooperation and individual participation. If we consider interaction as a form of action, then humans are not the only species capable of co-construction. Dogs, too, can engage in cooperative and participatory interactions, both within their own species and across species boundaries. In fact, humans and dogs appear able to hybridize their intraspecific communication systems, transforming them into mutually understandable, interspecific forms [2]. While multiple studies suggest that dogs can cooperate with humans [3,4,5], these investigations often exhibit limitations, including small sample sizes [3,4], minimal sociodemographic context [3,4], and a lack of attention to individual or shared (hybrid) experiences [3,4,5]. These gaps may contribute *to variability in how dyadic co-construction is expressed* [6]. Despite growing interest in cooperation and participation in human–dog interactions, there remains a lack of conceptual clarity regarding the very existence and definition of dyadic co-construction. *Does co-construction exist in owner–dog relationships? If so, how can it be defined, and what frameworks can help to conceptualize it?*

One promising lens is self-determination theory (SDT), a theory originally developed to understand human motivation and well-being [7]. SDT posits that individuals strive to fulfill three basic psychological needs—autonomy, competence, and relatedness (attachment)—and that satisfaction of these needs supports well-being and adaptive functioning; SDT classifies motivation along a self-continuum [7], ranging from:Intrinsic motivation: doing an activity for its inherent interest and enjoyment (most autonomous).Extrinsic motivation: driven by external outcomes, further divided into four types, (integrated, identified, introjected and external), from most to least autonomous.Amotivation: a state of disengagement and lack of intention, representing non autonomous regulation.

Autonomous forms of motivation tend to promote psychological resilience, adaptive behavior, and overall well-being [7]. These motivations emerge when individuals can satisfy the three fundamental needs:Autonomy: experiencing volition and self-direction.Competence: feeling effective and capable.Relatedness: feeling connected to others, accepted, and supported by peers.

When these needs are unmet or (actively) frustrated by the social environment, individuals may experience maladaptive outcomes [7].

While SDT was developed for humans, it has potential applications for non-human animals. Behaviorist perspectives emphasize external stimuli in shaping behavior [8], whereas evolutionary approaches highlight environmental pressures and fundamental needs as primary behavioral drivers [9]. SDT bridges these views by framing behavior as influenced by internal motivation and basic psychological needs, elements that may be relevant for animals, especially *companion species* like dogs and cats, who share human environments and social structures [10,11]. However, applying SDT to animals is complex. Animals differ from humans in their access to morality, rationality, legal rights, and cognitive communication tools [10,12]. Still, recent research supports the *“animalization”* of SDT, especially for companion animals living in anthropogenic environments where access to autonomy, competence, and relatedness is largely mediated by humans [10,11,12,13]. In such contexts, *human (ir)responsibility* can either support or hinder animals’ *ability to fulfill their psychological needs*, thereby affecting their *well-being*.

Considering these reflections, it becomes essential to ask whether SDT can truly be extended to dogs, and what the implications of such an extension would be for both owners and the dyadic relationship. Despite growing interest in animal SDT, few methodological tools exist to assess psychological and motivational needs across species in a systematic, bidirectional way [12]. Most studies remain anthropocentric or interpret animal behavior through human-centered lenses, perpetuating anthropomorphism and overlooking animal agency [11,14]. Although qualitative methods, such as behavioral observations, have proven valuable in studying participation and cooperation [3,4,5], it remains unclear whether current tools can adequately assess dyadic co-construction, especially in its relational, evolving, and bidirectional nature. From these reflections, we propose two key hypotheses:Dyadic relationships are co-constructed, with participation, cooperation, and SDT offering a viable framework for their conceptualization.Existing measurement approaches, although not originally designed to assess dyadic SDT, may serve as “proto-tools” that can be adapted and refined to study this relational process.

The primary objective of this narrative review was to conceptualize the co-construction of owner–dog interactions through the lens of SDT. The secondary objective was to examine existing methodological approaches that could be adapted to explore this process, and to highlight their respective limitations.

For reasons of clarity and readability, the term “dog owner” was chosen, even though it is more reductive than other concepts [15]. It refers to a basic human status in relation to companion animals, without presuming the perceptions of the animal [16] or suggesting a more favorable dyadic co-construction. Other terms such as “guardian”, “carer/caregiver/caretaker”, or even “parent” could be more appropriate.

This review was not intended to be exhaustive but rather to provide a conceptual and exploratory contribution to an underdeveloped field. The same structure was favored in each determinant of SDT, animated using questions. The latter were systematically followed by elements of response and avenues for reflection, thereby linking conceptual interrogations with constructive analysis.

## 2. Materials and Methods

This narrative review was developed using a comprehensive search strategy across several reference databases. Specialized databases in the social and natural sciences were consulted, including PsycINFO, PubMed, and CINAHL, alongside a multidisciplinary database (Web of Science). In addition, general search engines such as Google, Google Scholar, and Cairn were used to access complementary materials, including book chapters and gray literature. This combination of sources enabled a broad and integrated exploration of the key themes addressed in this review.

A thematic framework was developed to guide the search process. Two main thematic axes were identified; each associated with specific keyword combinations (see Table 1 and Table 2):The co-construction of dyadic interactions in relation to SDT;Tools and frameworks used to assess the fulfillment of basic psychological needs within dyadic contexts.

Searches were conducted using these keywords across all listed databases. To refine the selection, the following filters were applied:Title and abstract relevance;Publication date (<10 years), ensuring the narrative review to reflect the most recent methodological and conceptual advances in a still emerging field;Language (French/English), as these options cover most of the relevant scientific literature, as appreciated by the authors.

However, older references were retained when they were deemed foundational or conceptually essential.

All references were managed using EndNote™ software (version 20.6, Clarivate, PA, USA). The selection process was led by L.M., who performed an initial screening of titles and abstracts. Only references that were fully accessible, demonstrated methodological rigor and strong conceptual relevance, and aligned with the objectives of the review were retained for in-depth analysis. Given the conceptual and reflective orientation of the narrative review, conceptual and theoretical articles were prioritized alongside original research papers and systematic reviews. L.M. synthesized the literature using detailed reading notes and thematic summaries. A draft synthesis plan was created, and each section was written by L.M., then discussed and refined collaboratively with E.T. and B.L., following an iterative review process until consensus was achieved.

## 3. Competence and Education

### 3.1. Theoretical Objectives and SDT

In recent years, canine education has progressively shifted from a model centered on human dominance over dogs toward a more integrative approach. While human well-being remains a central objective, other goals—focusing on canine welfare, and even bilateral benefits—are now emerging [34]. In theory, these objectives include [35]:*Establishing, developing, adapting, and consolidating a foundational dyadic system of interspecific communication and interpretation;**Identifying and (de)valuing (in)appropriate behaviors through inter-species (de)motivational techniques.*

These objectives theoretically align with the development of dyadic SDT and its actors. The aim is to co-construct an evolving dialog, grounded in human, canine, and ultimately dyadic competence; all of which are dynamic and develop over time. Although this perspective emphasizes reciprocity, it must also acknowledge the asymmetry between dyadic actors. Domestication appears to have shaped dogs in such a way that cooperative and empathic leadership from their owners constitutes an important requirement for their health and well-being [33,36]. *Are these theoretical objectives achievable in practice?*

### 3.2. What Techniques for Achieving These Objectives: Links with SDT and Dyadic Impacts?

The most widely used training approaches are based on operant conditioning [35], a method that shapes canine behavior by teaching that actions have consequences. These techniques are divided into three categories, each aimed at increasing desirable behaviors and decreasing undesirable ones (noting that what constitutes a “desirable” behavior is often anthropocentrically defined) [35]:Aversive techniques, based on negative reinforcement and positive punishment. Respectively, these involve removing a stimulus following a desired behavior or adding a negative stimulus following an undesirable one (e.g., electric collars, choke chains, prong collars, physical/verbal/visual punishments).Reward-based (positive) techniques, focused on positive reinforcement—where a desirable behavior is followed by a reward, and an undesirable behavior by the removal of a positive stimulus (e.g., treats, toys, praise, clicker training).Mixed techniques, which combines both aversive and reward-based methods.

The choice of technique has a significant influence on both the dyad and its individual members. It directly impacts the co-construction of the dyad and the expression of SDT’s core principles. For example, compared to positive methods, aversive techniques are associated with:An increase in stress-related behaviors and postures in dog [27,38];A higher risk of canine aggression [19].

While the causal link between canine training methods and aggression remains unclear [31], canine aggression itself is a complex behavior influenced by the dog’s (social) environment, experiences, and individual characteristics (see [20] for more details).

Conversely, exploratory studies suggest that positive training methods are associated with:Increased frequency of dogs gazing at their owners during interaction [39];Greater sociability with strangers and higher activity levels during play [42];Improved focus and performance during professional tasks [41] and training [42].

Joint attention, facilitated by eye contact and physical touch, plays a vital role in both human and canine cognitive processing. For example, a study using non-invasive wireless electroencephalography (EEG) measured inter-cerebral neural coupling in human–dog dyads [26]. Results showed that:Neural coupling occurred in the frontal and parietal brain regions, associated with joint attention;Mutual gaze and physical touch independently enhanced this coupling, with their combination having a synergistic effect;This neural synchronization increased over time, plateauing by day seven.

These findings suggest that sensory cues shared between species, such as gaze, voice, and touch, are crucial for enhancing interspecific interaction quality and reciprocity. Both dogs and humans possess competencies in directing attention, interpreting social cues, and adjusting behaviors accordingly [2]. From a neural perspective, these signals, as well as their accumulation over time, appear to be important for enhancing the outcomes and quality of an interspecific interaction [26]. For example [2]:Dogs can understand certain words, tones, intentions, and human emotions;Humans can interpret canine vocalizations and emotional expressions.

Thus, dyadic interaction appears to be co-constructed through both neural activity and communicative competencies. However, for this process to function positively, each individual must have:Sufficient autonomy to explore the other;Opportunities for self-expression;The ability to accumulate and interpret signals over time.

Positive training methods appear to facilitate this process by promoting communicative competence and dynamic dyadic performance. In contrast, aversive methods impose rigid behavioral expectations, restricting exploration and self-expression. For instance, dogs trained with aversive methods look at their owners less frequently, potentially missing key social cues [32,39]. This could result in:The frustration of basic psychological needs, leading to extrinsic or amotivated perceptions of training, and possibly behavior problems that affect human well-being;A breakdown in canine communication that increases the risk of misunderstandings or accidents.

In such cases, neither the dyad nor its members fully benefit from the co-construction of shared competence.

### 3.3. Intra-/Interspecific Factors Influencing This Psychological Need?

Yes! Multiple intra- and interspecific factors influence how this need for competence is fulfilled. For instance, the coherence of communication (i.e., timing of reward, human posture, and the dog’s attention) affects learning success [25]. Key factors include:Human adaptation to canine signals: In a two-choice task, dogs tend to respond better to human gestures than to voice commands [18].Dog size: Owners of small dogs are less consistent in training and play than owners of large dogs, which correlates with lower obedience scores [37].Reward type: Food is generally more effective than petting, which itself outperforms verbal praise [29,40]. However, preferences may vary by familiarity, environment, body area touched, and individual canine personality [2,40].

Regarding canine traits, factors such as skull shape, breed, sex, age, and training level influence social perception and behavior [30,34]. For example [30,34]*:*Dogs exposed to humans early are more likely to understand pointing gestures;Older dogs may be less responsive to facial expressions;Brachycephalic dogs show more attention to human faces;Breed and work style influence gaze duration and task performance, although breed only accounts for a small portion of behavioral variation [23]. Behavioral expressions and predispositions are linked to a certain genetic selection but also to environment, utility, training, and experience.

Among humans, gender and personality also seem to influence training preferences. One study found that female trainers were more likely to use non-aversive methods and be certified [21]. No recent study has established a link between the choice of training method and the gender of the owner. Personality and mental health of the owner play a role as well. Owners high in agreeableness were less likely to use physical or verbal corrections [24]. Neuroticism in owner was associated with dog aggression [20]. A study based on *n =* 1564 dyads, very weak evidence linked owner personality and dog behavior, mediated using aversive training methods [19]. But moderate depression in male owners correlated with a fivefold increase in aversive method use [19]. Some studies also suggest that the owner’s experience with dog is important in canine education [22,43]. Other factors include *owner experience and cognitive abilities*. For example, success in canine certification correlated with high owner cognitive skills, training frequency, and low levels of perceived canine disobedience [28]. In the context of canine education, the cognitive abilities of owners are not necessarily optimized. In fact, a systematic review (*n =* 29) explains that canine education is often based on oral instructions and advice [17]. This method seems moderately effective in teaching owners the necessary competence to train their dogs. To remain effective, it should last between 3 and 6 months, which is not always the case [17].

However, many of these studies are based on small, non-representative samples and diverse methodologies, limiting generalizability. While they were not designed with SDT in mind, they collectively highlight the need to account for *individual and collective variability* in dyadic interactions [6], a topic relevant to the next sections.

### 3.4. Practical Feasibility of These Theoretical Objectives?

These objectives are not always achievable in practice. Their success depends on the choice of training techniques, and biopsychosocial, experiential, and environmental variables specific to each dyad. These elements are interdependent, making causality difficult to establish. To better understand this complexity, longitudinal studies are needed.

Despite these challenges, dyadic co-construction consistently occurs, though some combinations of factors lead to more successful outcomes. Effective dyads enable exploration, error-making, and iterative improvement. They build upon individual strengths and motivations, which merge into shared dyadic motivation, reinforcing competence.

**Key takeaways**:*Theoretical* vs. *practical gaps: While theoretically sound, these objectives are not always fully achievable in real-life contexts.**Impact of educational techniques: The choice of an educational technique influences the fluidity and stability of dyadic co-construction.**Effectiveness of positive methods: Reward-based methods tend to facilitate better fulfillment of fundamental psychological needs than aversive methods.**Need for further research: Future studies should explore the interplay of biopsychosocial, experiential, and environmental factors in shaping dyadic competence over time.*

## 4. Attachment

### 4.1. Can We Talk About Bilateral Relatedness Between Owners and Dogs?

In Western countries, dogs play an essential role in households. In 2024, the estimated number of dogs was 8.3 million in Canada [83], 9.9 million in France [84], and 89.7 million in the United States [85]. That same year, dogs ranked second to cats in pet ownership in Canada and France (8.9 million and 16.6 million, respectively) [83,84], while in the United States, dogs appeared to be the preferred companion animal, outnumbering cats (73.8 million) [85].

Across time, cultures, and societies, both (non-)Western, the role of companion animals—particularly dogs—has evolved and diversified across different functions [87]. Taking the example of Western European societies, it appears that with the rise in living standards, keeping a companion animal at home became increasingly popular [70,75,87]:In the Middle Ages and early modern times, moralists, especially within the discourse of the Catholic Church, condemned the practice of keeping animals solely for companionship. However, in wealthier households and religious institutions, their presence was already considered normal in practice and everyday life, and their role could also include that of companionship.In the XVI^th^ and XVII^th^ centuries, European aristocratic households kept a variety of species they considered companion animals, including dogs and exotic animals.In the XVIII^th^ century, the rise of the urban middle class contributed to the spread of companion animal ownership to broader segments of Western society. Factors behind this trend included the bourgeoisie’s desire to emulate the aristocracy and increased rural-to-urban migration.During the Victorian era, companion animal keeping became a family activity for the middle class. Companion animals were seen as having both social and educational benefits, especially for children.

Since the XIX^th^ century, companion animals have come to symbolize a *mutual relationship* [67], and many now consider them fully fledged *family members* [86]. From a SDT perspective, this evolution is particularly interesting. To fully understand the emergence, development, and current state of dyadic relationships, we must consider not only present-day interactions or recent life events but also cultural and historical dimensions. In this sense, traces of dyadic co-construction can likely be found as far back as the domestication of dogs.

From a theoretical standpoint, human attachment to companion animals is grounded in the same principles that govern infant–caregiver attachment [53,56,81]. Attachment theory [71] explains that are biologically predisposed to form close bonds with their caregivers, bond that can vary in quality and significantly affect development. In adults [57,68]:A high level of attachment anxiety refers to individuals who worry about the reliability or availability of those they are attached to.A high level of attachment avoidance refers to individuals who distrust or distance themselves from their attachment figures.Secure attachment, in contrast to these insecure styles, is characterized by confidence in the reliability and responsiveness of others, which promotes well-being.

Beyond attachment, the biophilia hypothesis suggests that humans are genetically and evolutionarily programmed to be connected with nature and animals [72,73]. The love of animals seems almost innate: It has been demonstrated that human infants (4 to 12 months old) show greater visual attention and emotional engagement toward various animals than toward objects [74].

*Do dogs form attachments to their owners?* Several lines of research suggest they do. In separation–reunion tests, dogs show behaviors indicative of attachment [58], including:seeking proximity to their owners;displaying varying levels of distress during separation;actively initiating contact upon reunion.

These behavioral observations are supported by neurological evidence. Functional MRI (fMRI) studies show that brain regions such as the caudate nucleus, hippocampus, and amygdala are activated when dogs view familiar, emotionally salient human faces [65]. These regions are associated with reward processing, memory, and emotion. Activation of the caudate nucleus, in particular, may reflect a learned association with human-mediated rewards [65] or general canine motivation [76]. Furthermore, dogs appear to develop attachment styles like those seen in human infants toward their caregivers [62,66]. This suggests that dyadic attachment involves not only human feelings and perceptions but also the canine brain, body, and emotions. *Is attachment within a dyad co-constructed by both the human and the dog?* (see below Section 4.4)

### 4.2. What Are the Impacts of This Fundamental Psychological Need on Dyads and Their Participants in Relation to SDT?

On the canine side, research, especially using the strange situation procedure (SSP), offers valuable insights. Compared to insecure attachment, secure attachment in dogs appears to have numerous benefits for both the dog and the dyad:Dogs with secure attachment increase exploratory behavior and proximity to humans, whether the human is a familiar owner or a stranger [61,66]. The effect of the owner, however, has a stronger effect [66].They play more with toys when the owner is present, but not with a stranger [46,77,78].They show reduced asymmetry in rightward tail wagging in the presence of a stranger when their owner is near [80].

The tail is a complex communication tool in dogs. Its asymmetry, position, amplitude, and wagging speed convey emotional states [2]. From a neurological perspective, tail wagging is influenced by cerebral lateralization. Positive stimuli, like seeing the owner, activate the left hemisphere, producing rightward wagging. Negative or withdrawal-related stimuli activate the right hemisphere, leading to leftward wagging [59,82].

*Does the owner’s attachment style affect canine behavior?* Yes. In a study involving 26 dogs with a history of aggression (13 toward humans and 13 toward other animals), Gobbo and Zupan found [20]:Dogs aggressive toward strangers tended to have owners with lower levels of anxious attachment;Dogs aggressive toward their owners had owners with higher avoidant attachment and lower conscientiousness.

From an SDT perspective, the first finding is intriguing—one might expect the opposite. *Could this be due to owners being overconfident in their dog, leading to reduced vigilance and allowing aggressive behaviors to go unchecked?* The second result is more intuitive: owners with strong avoidant attachment may struggle to meet their dog’s emotional needs, undermining the dog’s ability to see the owner a secure base [20], which can in turn lead to aggression. These findings support the idea of a co-construction process between human and canine attachment styles.

On the human side, a systematic review (*n =* 40) examining the link between attachment to companion animals and depression found [68]:most studies reported a positive or neutral correlation with depression;secure attachment was negatively associated with depression;insecure attachment (anxious or avoidant) was positively associated with depression;Still, causality remains hard to establish [68].

*What happens when we combine human and canine attachment data?* A systematic review (*n =* 6 studies) on attachment found that [44]:Owners with secure attachment often had dogs with secure attachment.Owners with avoidant attachment tended to substitute human relationships with canine ones, and their dogs also displayed avoidant or anxious styles.Owners with anxious attachment preferred turning to their dog during stress, and their dogs often developed secure attachment.

In sum, insecure attachment, whether human or canine, can be detrimental to dyads and their fulfillment of SDT. Secure attachment, by contrast, fosters exploration, interaction, and psychological well-being. Owners with avoidant attachment appear to pose the greatest risk for insecure canine attachment and dysfunctional dyads [44]. In such cases, dogs may struggle to find in their owner a reliable attachment figure to refer to confidently [20,58].

*What about grief?* Most pet owners experience significant grief after a companion animal’s death. This sadness is unique to each individual and may be associated with anxiety, stress, shame, or trauma [69]. A narrative synthesis (*n =* 36) showed that maintaining emotional bond with deceased companion animal can intensify grief [69].

Dogs also appear to experience grief. In triads (an owner and two dogs) when one dog dies, the surviving dog exhibit fear or behavioral changes [63]. Predictors of this included [63]:The quality of the bond between the dogs (friendly or parental; shared resources);The emotional state of the grieving owner.

Interestingly, the length of the relationship did not influence these behaviors (in the surviving dog), only its intensity. This suggests that dyadic attachment persists beyond physical presence, requiring eventual deconstruction and emotional acceptance by both human and dog.

### 4.3. Is This Psychological Need Influenced by Intra-/Interspecific Factors?

Attachment strength is influenced by many factors [44]. On the dyadic level, interpersonal dyadic complementarity and relationship length are key [44,47,48,49,50,51,64]. The quality and content of shared activities also matter. Gender and familiarity may also shape interactions. In one SSP study, shelter dogs showed more concern toward unfamiliar men than women, possibly due to prior socialization patterns [79]. Similarly, another study found:Male shelter dogs accompanied by unfamiliar men were more likely to adopt juvenile urinary postures and urinate less;When walked by unfamiliar woman, dogs defecated less [55];Female dogs showed similar patterns for defecation behaviors [55].

On the canine side, variables like age, breed, and dyadic history affect attachment [47,48,49,50,51,64]. On the human side, attachment strength is positively associated with being a woman [45,52,54,57]; having an agreeable personality; not having children; being the sole caregiver, positive interaction history and companion animal perception [44].

### 4.4. Is Attachment Within a Dyad Co-Constructed by Both the Human and the Dog?

Probably. However, most existing studies focused more on human attachment to dogs than the reverse [60]. Research is also limited in cultural, social, and environmental diversity, and often suffers from small samples and a lack of breed variety [47,48,49,50]. Despite these limitations, findings on secure attachment hint at *benefits for both species’ psychological needs, including autonomy, relatedness, and competence*. However, this connection has not yet been explicitly studied. Future research should explore:Dog’s attachment to humans in more depth;Influences of biopsychosocial factors, lived experiences, and environmental contexts;Longitudinal effects on dyadic development.


**Key takeaways:**

*Bilateral attachment: Attachment should no longer be viewed as a one-sided human experience, it is co-constructed.*

*Attachment style and co-construction: Secure attachment fosters more positive dyadic co-construction than insecure styles.*

*The example of grief: In the face of difficult events, humans and/or dogs seem to enter processes of deconstruction of the previously experienced co-construction and of emotional acceptance.*

*Need for further research: Future studies must deepen the understanding of canine attachment as well as the various factors that shape dyads across time and space. Beyond the experience of grief, when faced with life’s challenges, are dyads capable of undergoing deco-construction and then reco-construction?*



## 5. Autonomy

### 5.1. Theoretical Objectives and SDT

Just as with the literature on attachment, that which addresses dyadic autonomy is incomplete (Table 2). As defined within the framework of SDT, autonomy has only recently attracted scholarly interest [13]. Available data suggest that autonomy should aim to:*Allow dogs to satisfy both their psychological and physiological needs while balancing the needs of both human and dog* [89,93].*Respect canine choices and (dis)satisfactions* [12,88].*Promote a safe environment for all members of the dyad* [12].

These are theoretical objectives. They mark a departure from cartesian perspectives and human exceptionalism, promoting instead deeper recognition of canine identity [88]. Dogs are no longer regarded solely as individuals, but as beings whose identity must be respected for their well-being. Moreover, humans tend to perceive them as “persons” and to seek to understand them through anthropomorphism [33,36]. Although this interpretative approach is often criticized, Kotrschal proposes the idea of an enlightened anthropomorphism, supported by measured empathy [33]. Such an approach could foster a more harmonious co-construction, helping humans to respond more appropriately to their dog’s needs while allowing the latter to express themselves as an active social partner.

Thus, at the heart of this perspective lies the co-constructed balance between canine, human, and dyadic autonomy. *To what extent are the theoretical objectives of this psychological need practically achievable?*

### 5.2. What Techniques for Achieving These Objectives?

#### 5.2.1. Relationship with the Physical and Social Environment: SDT-Relevant Influences (Impacts and Intra/Interspecific Factors)

Canine autonomy is closely tied to the quality (understood in canine terms) of the artificial environment in which dogs live. “Caninizing” the human environment allows dogs to engage in natural behaviors. Enrichment benefits not only dogs but also humans and their shared relationship. It can help regulate (un)desirable behaviors [97] and potentially support healthy aging [98]. According to SDT, and as discussed by Jones [88], the physical and social environment shapes the choices available to a dog. A highly enriched space supports autonomy by offering genuine opportunities for choice and self-expression. Conversely, overly restrictive or under-stimulating environments limit autonomy and can contribute to frustration or stress. The number, nature, and perceived value of choices matter. For instance, if a dog is offered only two options that hold little intrinsic interest, its experience of autonomy is likely to be diminished compared to having five options, including two that align with its preferences. As with the elephants studied by Bell Rizzolo and Bradshaw [14], dogs require a human partner who is genuinely invested in meeting their needs. Co-construction can be observed through:Analysis of the physical and social environment provided to the dog.The extent to which the dog appropriates this environment.The owners’ understanding of and responsiveness to their dog’s preferences.

Dyadic performance can also reflect autonomy. In an observational study of 12 military avalanche search teams, dogs who performed best in simulated trials appeared more autonomous and less handler-dependent than those who failed [91]. Successful dogs spent:More time sniffing/exploring (*p <* 0.001) and digging intensively (*p <* 0.05).Less time seeking proximity or contact with their handler.

Handlers’ attentiveness also improved canine performance [91]. It is possible that these effects are less noticeable in dyads composed of ordinary owners and companion animals. However, companion dogs appear to regulate their behavior based on their owners’ emotional cues. They approach or move away from an object based on the positive or negative emotional message received from their owner, but do not react the same way to strangers’ message [99,100,101]. Autonomy, therefore, is not equivalent to independence or detachment.

These findings suggest that canine autonomy is co-constructed through individual characteristics, education, relational strength, and dyadic communication. Dogs that appear more autonomous may have internalized the actions they are performing, indicating intrinsic motivation and a sense of control over their body and environment. They may feel more comfortable exploring and successfully finding a solution to the given problem [88].

#### 5.2.2. Negotiating Canine Consent: SDT-Relevant Influences

Dogs are sentient, emotional beings capable of reasoning [95,103]. This capacity implies a form of consent [11], whereby dogs can choose to participate, cooperate, or express their (dis)likes [88]. For this consent to be valid, human partners must perceive and respect it [88]. *Can canine consent improve the course of a dyadic interaction? Does it seem to contribute to its co-construction?* Cooperative care is a valuable model here [88]. Its goal is to allow dogs to exert control and actively participate in their care. For this, the dyad must develop a shared language that supports both the expression and interpretation of canine (non)consent. Through routine, redundancy, and mutual anticipation, both dog and human can better navigate interactions. From the SDT perspective, this exemplifies co-constructed autonomy in care contexts. Since each party understands what to expect, engagement, cooperation, and consent become easier. However, this dynamic can be undermined by relational quality, trust, comfort, and mutual knowledge [88]. The owner’s knowledge and willingness are also important [88]. Many humans struggle to correctly detect or interpret canine signals [11,94]; others take the excuse of canine autonomy to not assume their leadership, therefore dogs could take over and start controlling (as children) their parent(s) to gain security; and some consciously ignore them [11]. Environmental and contextual factors, including interaction duration and content, also influence a dog’s stress, and by extension, its autonomy [94].

#### 5.2.3. Perceptions of Human Autonomy in a Dyadic Context: SDT-Relevant Influences

A dog’s illness or behavioral problems may reduce both canine autonomy and human autonomy, limiting owners’ ability to pursue daily activities, interests, or relationships [89,93]. A study of 1693 dyads found that many owners perceived meeting their dogs’ needs as burdensome [93], which impinges on their own autonomy. Since dogs depend entirely on humans, the latter bear full responsibility for fulfilling their needs [90].

The autonomy of human actors also depends on dyadic activities. For instance, recreational walks are perceived positively, while functional walks often feel like obligations, associated with guilt or duty rather than pleasure [89,93]. Regulations can also restrict autonomy. For example, permanent leash use is imposed in some jurisdictions, which may frustrate both humans and dogs [92,102]. Co-construction thus extends beyond the dyad. Public policies, legal frameworks, and various stakeholders also play roles while altering the predictability of dogs relationship with their parents (see [96]).

### 5.3. Practical Feasibility of These Theoretical Objectives?

In reality, dogs rarely experience full autonomy. What they experience might better be described as “artificial” or “perceived” autonomy [88]. The goal, therefore, is not independence but interdependence, an optimal, co-constructed balance over time and space [13]. Within this dynamic, both human and canine partners should ideally experience mutually respected freedom of action and expression. However, not all dyads achieve, or are even capable of achieving, such balance. As with education or competencies, dyadic autonomy depends on a complex interplay of biological, psychological, social, and environmental factors.


**Key takeaways:**
*Theoretical* vs. *practical tension: Despite its idealistic aims, the current literature remains limited.*
*Dyadic interdependence: Dogs do not possess absolute autonomy. Instead, autonomy emerges through co-constructed interdependence with humans.*

*Research gaps: Further studies are required to explore how human, canine, and dyadic actors relate to these theoretical goals. Future work should consider biopsychosocial factors, life experiences, and physical/social environments across timeframes.*



## 6. Researching and Measuring Co-Construction

### 6.1. Engaging, Observing, Questioning

The preceding sections indicate that dyadic co-construction can be approached with both qualitative and quantitative methodologies. Each family of methods has distinct strengths and limitations.

Interviews and open-ended questionnaires (commonly used to study human autonomy in dyads) are particularly powerful for accessing owners’ lived experience and contextualizing behaviors [89,90,93]. Their main strength lies in depth: they allow deconstruction of practices, meanings, and intentions. Their principal weakness is that they provide limited quantification, preventing the assignment of confidence weights, generalization, or systematic comparison across large samples.

Direct observation and standardized behavioral tests yield precise, controlled, and comparatively objective data [91,97]. Examples include task-based protocols [91,117] and adapted versions of SSP to assess attachment [66,107]. These methods excel at detailed, behavior-level analysis but are resource-intensive: they require trained observers, can be costly and time-consuming, and typically cannot be scaled easily to large, representative samples. They also often remove dyads from their natural context and are susceptible to reactivity and situational biases [114].

Taken together, these approaches underscore a key issue: studying dyadic SDT without situating findings within the actors’ individual and shared experiences risks producing fragmentary or non-comparable results. Many studies provide sparse sample descriptions and use heterogeneous protocols, which complicates synthesis and weakens external validity. *Under such conditions, co-construction can only be explored as a snapshot rather than as an evolving, context-dependent process*.

### 6.2. Exploration of the Nervous System

Neurophysiological tools (EEG, fMRI) and autonomic measures (ECG/heart-rate variability) offer complementary, objective windows into neural and physiological correlates of interaction, learning, emotion, and attachment [119]. fMRI captures hemodynamic responses reflective of regional brain activity; EEG indexes electrical activity with high temporal resolution; ECG-derived heart-rate variability (HRV) indexes autonomic balance and stress responsivity.

Recent systematic reviews summarize the scope of canine neurophysiological research: EEG studies (*n =* 22) have examined sleep, learning, sensory perception, language processing, social cognition, and emotions [118], while fMRI studies (*n =* 46) have mapped canine brain responses to resting-state and multi-sensory stimuli [105]. Studies integrating human and canine measures (e.g., simultaneous recordings) highlight reciprocal neural coupling during interaction and are promising for studying co-construction [26]. They enable the exploration, from a neurological perspective, of the well-being expression [104,124] and/or attachment [112,115] that occurs between a human and a dog. However, several methodological limitations constrain interpretation and generalizability: heterogeneous and often small samples with occasional dataset overlap [112,115]; variability in EEG electrode types, placement, and preprocessing [118]; sensitivity of HRV to context and individual physiology [104]; electrical signal baseline drifts, and electrical interferences; movement and muscle artifacts in awake recordings [120]; and inconsistent handling of anesthesia effects in fMRI/EEG studies (some studies include anesthetized subjects, others do not) [105]. Training dogs to tolerate equipment greatly improves data quality, but training protocols are inconsistent and underreported [105].

Notably, co-construction may extend into unconscious or off-task states (e.g., sleep). EEG research links sleep parameters to learning, memory consolidation, and social functioning in dogs [118]; actigraphy and dyadic co-sleep studies offer a lower-cost avenue to explore nocturnal or unconscious aspects of dyadic life [106,108,110,116,123].

### 6.3. Measuring (Psychometrics and Behavioral Scales)

Psychometric instruments have proliferated for human–animal interaction research. They are scalable, relatively inexpensive, and amenable to remote administration, which supports larger, more generalizable samples. Typical instruments target owner reports as proxies for dyadic states [114]. Here are some examples of instruments used in dyadic research:Dog impulsivity assessment scale (DIAS) [121]: 18 items addressing behavioral regulation, aggression, and reactivity (5-point Likert). Shows promising validity; reliability needs further demonstration [121,122].Canine behavioral assessment & research questionnaire (C-BARQ): extensive (≈101 items) owner-reported measure covering multiple behavior domains, with established reliability and partial validity [114].Pet attachment questionnaire (PAQ) [81]: measures owner attachment (to a companion animal) style. Out of 26 items, testing anxious and avoidant attachment, 7-point Likert scale each, with acceptable reliability and partial validity.Lexington attachment to pets scale (LAPS) [125]: 23 items across factors of general attachment, substitution, and animal rights/welfare (4-point Likert scale), widely used in the literature.

Crucially, *no validated, widely accepted scale yet measures dogs’ attachment to owners*; instruments such as the dog attachment insecurity screening inventory are still under development [109]. It is intended to assess the canine attachment style to its caregiver. Likewise, *there is no validated scale that directly quantifies dyadic autonomy as a bilateral construct, and few instruments adequately capture canine motivation* (though the WDC-BARQ explores working-dog motivations [113].

Limitations of owner-report scales include *respondent bias, item misinterpretation, and the unilateral nature of the perspective: most scales capture what owners observe or feel, not the dog’s experience*. Consequently, apparent behaviors (e.g., impulsivity on DIAS or problem behaviors on C-BARQ) may reflect unmet psychological needs, stress, or constrained autonomy rather than stable trait pathology [111,126]. Without concurrent canine-centered measures, interpretation remains ambiguous.

### 6.4. Lessons from “Proto-Tools” and Paths Forward

A variety of methodological approaches currently serve as starting points for documenting dyadic co-construction. These include subjective interviews and questionnaires, behavioral observations and standardized tests, neurophysiological methods, and multidimensional psychometric scales. However, they all have advantages and limitations. Qualitative methods provide depth and contextual richness but lack quantification and are difficult to generalize. Behavioral observations and standardized tests offer objective, fine-grained data but are resource heavy, often small-sample, and may disrupt natural contexts. Neurophysiological measures can objectify underlying processes (e.g., neural synchrony, autonomic regulation) but suffer from methodological heterogeneity, small samples, artifacts, and variable training/anesthesia protocols. Multidimensional psychometric scales are scalable and standardized but commonly adopt a unilateral (owner-centered) perspective and may miss canine agency and motivation.

Recommendations to advance measurements of co-construction:*Develop bilateral instruments that simultaneously integrate owner and canine perspectives (*e.g.*, matched owner-report and behavioral/physiological indices; dyadic questionnaires that capture shared routines, negotiated choices, and mutual influence).**Adopt mixed-methods designs combining qualitative context, standardized behavioral probes, scalable psychometrics, and targeted neurophysiological measures. This triangulation would leverage the strengths of each approach.**Standardize protocols for EEG/fMRI/HRV in dogs (e.g., electrode placement, preprocessing, training procedures, and anesthesia handling) to improve comparability and reproducibility.**Prioritize longitudinal and naturalistic studies to capture co-construction as an evolving process and to disentangle causal pathways among technique, experience, environment, and dyadic outcomes.**Report samples and contexts rigorously (e.g., breed, age, training history, owner demographics, environment) to allow appropriate stratification and meta-analytic synthesis.*

In short, current “proto-tools” can document elements of dyadic co-construction but remain largely unilateral and fragmented. Progress requires integrative, bidirectional, longitudinal approaches that respect the dyad as the primary unit of analysis.

## 7. Conclusions

From an SDT perspective, dyadic co-construction between humans and dogs is far more than simple participation or cooperation. It represents a dynamic, evolving process of interaction and mutual adaptation, in which each partner influences, and is influenced by, the other (Figure 1). At a minimum, these exchanges aim to meet each individual’s fundamental psychological needs, and in some cases, to actively support the partner’s needs as well. This perspective rejects unilateral control: interactions arise from a synergy shaped by bio-psycho-social, experiential, and environmental factors, with both actors contributing to the structure and evolution of the dyad.

Methodologically, dyadic co-construction can be examined through various approaches. Qualitative tools capture rich, detailed interactional snapshots but lack statistical generalizability. Mixed-method designs may help overcome this limitation, yet existing scales tend to adopt a unilateral lens. Advancing this field could benefit from techniques that explore nervous system activity, which may provide objective evidence of patterns already qualitatively described, and thus foster more balanced, bidirectional assessments. To date, however, no validated tool fully captures dyadic co-construction from both human and canine perspectives.

Although physiological and molecular biomarkers (e.g., heart rate, blood pressure, cortisol) are widely used in research related to SDT, health, and well-being, they were excluded from this review because they provide a more distal perspective on co-construction and yield inconsistent findings [104,127,128]. For instance, a systematic scoping review (*n =* 27) of animal-assisted interventions (AAI) [128] found that, while most human-focused studies (*n =* 18) reported at least one positive stress-related outcome, correlations between biomarkers and physiological measures were absent, and results between biomarkers and self-reported stress were inconsistent. Canine-focused studies (*n =* 9) also produced inconclusive results, particularly for non-certified dogs. Only three studies combined physiological and molecular markers, preventing meaningful correlation analyses

Practical challenges further limit biomarker use. Cortisol, for example, follows a circadian rhythm and is affected by physical activity, diet, and sampling methodology—variables that are often underreported [128]. In dogs [127], cortisol concentrations can also vary by age, sex, reproductive status, and even by the presence of the owner, yet socio-demographic details are rarely documented [128]. Moreover, to be optimal, cortisol levels interpretation needs parallel behavioral observations/measurements, considering its lack of specificity as biomarker for stress, anxiety, pain, or … positive engagement [128]. Interestingly, the greater cortisol modulation in interspecies social situations in dogs as compared to socialized wolves ultimately seems to serve the ability to adapt to the human-dominated environment—one of the essential results of *dogification* (domestication) [129,130]. This constitutes an underlying mechanism of dyadic co-construction.

This narrative review suggests that the relationship between dogs and their owners may be shaped by a dynamic process of co-construction, in which each partner continuously adapts to the other. Applying SDT to this context offers a valuable framework for understanding how the fundamental psychological needs of autonomy, competence, and attachment can be mutually fulfilled, or hindered, within the dyad. Co-construction appears to be influenced by a complex interplay of biopsychosocial factors, individual and shared experiences, and environmental conditions. While current methodological approaches provide useful insights, they fall short of capturing the bidirectional and evolving nature of this process in a standardized, statistically robust way. Advancing the study of dyadic co-construction will require the development of integrative, longitudinal, and multidimensional tools capable of simultaneously considering human and canine perspectives. Such advancements could contribute to evidence-based practices that enhance both canine welfare and the quality of human–dog relationships.

## Figures and Tables

**Figure 1 animals-15-02875-f001:**
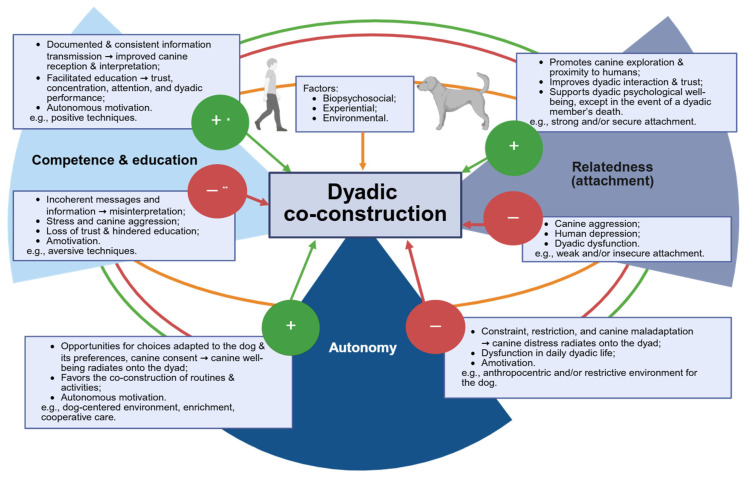
Synthesis of dyadic co-construction in perspective with self-determination theory ***. This figure illustrates the process of dyadic co-construction through the three basic psychological needs associated with Self-Determination Theory. Each need is influenced by biopsychosocial, experiential, and environmental factors specific to humans, dogs, and dyads. Their satisfaction or frustration can generate effects that influence dyadic motivation, functioning, and well-being. Together, these dynamics determine the quality and stability of dyadic co-construction. * Positive expressions related to the satisfaction of the basic psychological need. ** Negative expressions related to the difficulties in satisfying the basic psychological need. *** Created in BioRender. Martin, L. (2025) https://BioRender.com/q9ncsm5, accessed on 28 September 2025.

**Table 1 animals-15-02875-t001:** Pathways used in each reference bank according to main theme.

Main Theme	Path Used
Co-construction	(co-construction OR self-determination OR motivation OR partner OR coopera*^1^) AND (interspeci* OR animal OR dog OR canine)
Competence & education	(education OR training OR behav* OR cognit* OR competence OR communication OR reinforcement OR avers* OR interaction) AND (dog OR canine) AND (owner OR human OR interspeci*)
Relatedness (attachment)	(attachment OR bond OR relationship OR affiliation OR synchronization OR referenc* OR secur*) AND (dog OR canine) AND (owner OR human OR interspeci*)
Autonomy	(autonomy OR consent OR liberty OR desire OR activit* OR environment* OR norm* OR regulation) AND (dog OR canine) AND (owner OR human OR interspeci*)
Researching and measuring co-construction	(co-construction OR self-determination OR motivation OR partner OR coopera* OR education OR training OR reinforcement OR avers* OR interaction OR autonomy OR consent OR liberty OR desire OR activit* OR environment* OR norm* OR regulation OR attachment OR bond OR relationship OR affiliation OR synchronization OR referenc* OR secur*) AND (dog OR canine) AND (owner OR human OR interspeci*) AND (tool OR scale OR metric OR measure OR assessment OR observation OR marker OR physi* OR neuro* OR actigraphy OR perform* OR method* OR brain OR behav* OR cognit* OR competence OR communication)

^1^ The use of asterisks in the keyword search strings made it possible to automatically retrieve all derived forms of a term from its root.

**Table 2 animals-15-02875-t002:** Number of references retrieved for and used in each main theme (13 February 2025) ^1^.

Main Theme	Number of References Drawn from Databases with Intra-Database Duplicates (*n* =)	Number of References Drawn from Databases Without Intra-Database Duplicates (*n* =)	Number of Cited References (*n =*) ^2^
Co-construction	*n =* 19,613	*n =* 14,184	*References found from the script:* Original research articles: *n =* 3 [3,4,5] Others: *n =* 3 [10,13,14] *Additional references**:*** Books: *n =* 3 [1,7,12] Others: *n =* 3 [2,6,11]
Competence & education	*n =* 12,252	*n =* 8766	*References found from the script:* Systematic review: *n =* 1 [17] Original research articles: *n =* 12 [18,19,20,21,22,23,24,25,26,27,28,29] Others: *n =* 6 [2,6,30,31,32,33] *Additional references**:*** Books: *n =* 3 [34,35,36] Original research articles: *n =* 7 [37,38,39,40,41,42,43]
Relatedness (attachment)	*n =* 7041	*n =* 5129	*References found from the script:* Systematic reviews: *n =* 2 [44,45] Original research articles: *n =* 22 [20,46,47,48,49,50,51,52,53,54,55,56,57,58,59,60,61,62,63,64,65,66] Others: *n =* 2 [2,67] *Additional references**:*** Systematic reviews or systematic narrative synthesis: *n =* 2 [68,69] Books: *n =* 6 [70,71,72,73,74,75] Original research articles: *n =* 7 [76,77,78,79,80,81,82] Others: *n =* 5 [83,84,85,86,87]
Autonomy	*n =* 14,534	*n =* 10,102	*References found from the script:* Book: *n =* 1 [88] Original research articles*: n =* 5 [89,90,91,92,93] Others: *n =* 3 [11,13,94] *Additional references**:*** Books: *n =* 4 [12,36,95,96] Original research articles*: n =* 6 [97,98,99,100,101,102] Others: *n =* 3 [14,33,103]
Researching and measuring co-construction	*n =* 16,199	*n =* 11,749	*References found from the script:* Systematic reviews, systematic literature reviews or meta-analysis: *n =* 2 [104,105] Original research articles: *n =* 17 [26,66,89,90,91,93,106,107,108,109,110,111,112,113,114,115,116] Other: *n =* 1 [117] *Additional references**:*** Systematic review: *n =* 1 [118] Books: *n =* 2 [119,120] Original research articles: *n =* 6 [81,121,122,123,124,125] Other: *n =* 1 [126]

^1^ The main discussion was based on *n =* 122 references, specifically: *n =* 8 systematic reviews, systematic narrative syntheses, and meta-analyses; *n =* 77 original research articles; *n* = 16 review articles and conceptual analyses; *n =* 1 research protocol; *n =* 17 books; and *n =* 3 websites. Additionally, *n =* 8 references—namely, *n =* 3 systematic reviews and a systematic scoping review, *n =* 2 books, *n =* 2 original research article and *n =* 1 other—are unclassifiable [8,9,15,16,127,128,129,130]. They were included for contextual purposes. ^2^ If a reference was cited in multiple sections, it appears in all relevant sections.

## Data Availability

Not applicable.

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
