# Peer review of "A Conceptual Framework for the Co-Construction of Human–Dog Dyadic Relationship"

_animals, 2025, doi:10.3390/ani15192875_

Round 1
Reviewer 1 Report
Comments and Suggestions for Authors
This narrative review examines how self-determination theory can enhance our understanding of co-construction between human-dog dyads. Overall, this is a well-constructed article and solid contribution to the literature base. Some clarifications are required throughout – please see comments below.
Feedback
- The concept of dog “ownership” has been challenged. While there are legal implications to the term “dog owner”, some have suggested use of alternatives such as “dog guardian”, “dog carer/caregiver”, “dog parent”, “canine guardian”, “dog custodian”, “pet caregiver”, and “dog steward” to honour the non-human animal and human bond, duty of care legislation, and animal sentience. Your article would benefit from inclusion of general commentary around this and provision of a justification for your use of the term “dog owner” and others of a similar nature.
- Lines 60-62: Are the following your research questions?: Does co-construction exist in 60 owner-dog relationships? If so, how can it be defined, and what frameworks can help to conceptualize it? They seem to appear suddenly without clear context or rationale.
- Lines 101-102: Are the following meant as a subheading?: Can dogs access SDT? If so, how? And what are the implications for their owners and the dyadic relationship as a whole? They seem to appear suddenly without clear context or rationale.
- What is your rationale for only examining articles from the past 10 years in this domain?
- What is your rationale for only examining sources in French or English?
- Lines 175-176: “Are these theoretical objectives achievable in practice?” I’m starting to see a pattern with your inclusion of questions throughout your document. Unfortunately, they do not have enough context for me to fully understand their purpose in each section.
- Section 3: Competence and education – I’m having trouble understanding how you got all this content based on your search terms, which do not include dog, canine, training techniques, etc. I have the same comment for all your “results” and narrative synthesis. Can you please clarify this in your methods?
- Line 259 – is it sex, or gender? Is there something linked to ones biology as male or female, or their socialization as men or women that influences training preferences?
- Do you believe this model could be applied to human-service dog or human-therapy dog dynamics?
Author Response
Please, refer to the attached file, including both reviewers and editorial board requests.

Reviewer 2 Report
Comments and Suggestions for Authors
General
In their review of dyadic co-construction in the frame of self-determination theory (SDT) the authors provide a real fresh and nicely comprehensive view of dog-human partnership and also make quite a number of concrete suggestions for further research (quantitatively and qualitatively by far exceeding the usual conclusion “more research is urgently needed” ).
For an evolutionarily orientated biologist (me), the problem with SDT is that this is a psychological construct. Although it may be part of the evolved human mindset, it is not a “natural feature” (i.e. one, which can be analysed on all four “Tinbergian levels” sensu Tinbergen 1963) - which does not mean that it may not be useful in the present context. Actually it is, in a pretty convincing way. One major problem with SDT in the present context is, that it remains unclear whether this construct is genuinely applicable to humans or whether it can be stretched to other animals (e.g. because of the Darwinian continuum). However, the authors succeed to argue that SDT may also be relevant at least for dogs, but their arguments could be even made more convincing (below).
Writing style and text organization make excellent reading, Fig. 1 is comprehensive and informative and most of the huge number of citations (gratulations not only for the systematic search, but also for integrating such a big amount of information into an informative review) is relevant. Not surprisingly , still some more relevant research (not cited) came to my mind (below)
A few specific comments
Ln 95: Elephant example: These is tamed Wildlife, not domesticated animals, therefore a truly suboptimal example of “companion animal”
Ln 172: “Ideally, this dialogue occurs without a hierarchical leader, allowing for a degree of autonomy on both sides. Each party must be able to express themselves, be understood (and understand), and argue to establish future rules. Are these theoretical objectives achievable in practice?” This sounds very much like anthropocentric ideology. It has been shown that while socialized wolves cooperate with humans at level, one of the consequences of domestication was, that companion dogs need socially competent leadership from side of their humans – also as a basic requirement for their wellbeing (below). This does not mean domination, nor is meant to downplay the autonomous decision competence of dogs. But human-dog dyads are basically asymmetric, as you discuss in your text, and domestication has made dogs fond of socially competent leadership by their human partners (below).
From ln 308: A VERY Euro- or even anglocentric historical perspective. In diverse cultures worldwide dogs have played diverse roles since the Palaeolithic .. and this has left traces in immigration societies such as the US or Canada.
Ln 309: “In medieval and early modern times, pet ownership was generally frowned upon. Moralists from the Middle Ages and Renaissance condemned keeping animals solely for companionship.” This was catholic church position at times, but practically and in everyday life, cats and dogs were kept during these time (even in monasteries) for functional reasons and also for companionship, particularly by the rich and noble ever since, since Roman times and much earlier. And there are numerous medieval paintings, Renaissance portraits with dogs (etc.)
Ln 327: “From a theoretical standpoint, human attachment to pets is grounded in the same principles that govern infant–caregiver attachment [51,54,76].” True, but not sufficient. Why are humans of virtually all ethnicities and times fond of companion animals (I really dislike the term “pet” for obvious reasons) at all? “Attachment” is a mechanistic principle (in the sense of Tinbergen 1963), which provides no insight at the evolutionary level. Towards this it would be useful, to integrate the concept of “BIOPHILIA” (sensu Fromm, Wilson, Kellert, etc.…) as a specific component of the typically human mindset, to be “instinctively” interested in nature and animals (see, for example the work of de Loache et al. on the the attention spans of babies a few months old).
Ln 363: “They show reduced asymmetry in rightward tail wagging in the presence of a stranger when their owner is near [64].” If I am not mistaken, there is nothing about the direction of tailwagging in Solomon et al. [64].
Ln 416: “Attachment strength is influenced..” In the basic attachment literature “strength” is no factor, but quality (which is, of course, debatable).
Ln 440ff: Takeaways .. not really new, see [64]
Ln 459: Dogs (as well as many other animals) are “persons” - according to most common definitions. Please see also: Kotrschal, K. Wolf–Dog–Human: Companionship Based on Common Social Tools. Animals 2023, 13, 2729. https://doi.org/10.3390/ani13172729
Ln 514: “Many humans struggle to correctly detect or interpret canine signals [11,88]” .. or are socially insecure how to respond, I would add. Optimal dog autonomy needs a socially competent leadership style on side of the human partner. It can be experienced quite often is that people are unable/reluctant to provide adequate leadership, being blocked by their weariness whether this would unduly restrict dog autonomy. In extreme (but still frequent) such cases, dogs take over and start controlling their owners to gain security, just as what children with disorganized attachment resort to as their only option to gain some predictability in their relationship with their parents.
Ln 686; Cortisol: The major reason for the inconsistent results with this nowadays easily measured and hence, abundantly used biomarker is mainly, that (salivary or systemic) cortisol levels need parallel behavioural observations/measurements for interpretation. Simply measuring the activation of the HPA axis by cortisol levels necessarily remains inconclusive as to what these levels mean. Identical cortisol levels can be caused by anxiety or by positive engagement. And in a wider perspective, cortisol levels are generally higher in dogs than in equally raised and kept wolves (please see: Vasconcellos AdS, Virányi Z, Range F, Ades C, Scheidegger JK, Möstl E, et al. (2016) Training Reduces Stress in Human-Socialised Wolves to the Same Degree as in Dogs. PLoS ONE 11(9): e0162389. doi:10.1371/journal.pone.0162389 and also: Jean-Joseph H, Kortekaas K, Range F and Kotrschal K (2020) Context-Specific Arousal During Resting in Wolves and Dogs: Effects of Domestication? Front. Psychol. 11:568199.doi: 10.3389/fpsyg.2020.568199). This has quite some relevance for dyadic co-construction, as the greater cortisol modulation in interspecies social situations in dogs as compared to socialized wolves ultimately seems to serve the ability to adapt to the human-dominated environment – one of the essential results of dogification (domestication).
Author Response

(The authors gave the same response as above.)
